# Identification and functional analysis of the CorA/MGT/MRS2-type magnesium transporter in banana

**MengYing Tong, Wen Liu, HongSu He, HaiYan Hu, YuanHao Ding, Xinguo Li, JiaQuan Huang[ORCID]\*, LiYan Yin**

Hainan Key Laboratory for Sustainable Utilization of Tropical Bioresource, College of Tropical Crops, Hainan University, Haikou, China

\* jqhuang@hainanu.edu.cn

## Abstract

Magnesium (Mg) plays an irreplaceable role in plant growth and development. Mg transporters, especially CorA/MGT/MRS2 family proteins, played a vital role in regulating Mg content in plant cells. Although extensive work has been conducted in model crops, such as Arabidopsis, rice, and maize, the relevant information is scarce in tropical crops. In this study, 10 *MaMRS2* genes in banana (*Musa acuminata*) were isolated from its genome and classified into five distinct clades. The putative physiochemical properties, chromosome location, gene structure, cis-acting elements, and duplication relationships in between these members were analyzed. Complementary experiments revealed that three *MaMRS2* gene members (*MaMRS2-1*, *MaMRS2-4*, *MaMRS2-7*), from three distinct phylogenetic branches, were capable of restoring the function of Mg transport in *Salmonella typhimurium* mutants. Semi-quantitative RT-PCR showed that *MaMRS2* genes were differentially expressed in banana cultivar 'Baxijiao' (*Musa spp. AAA Cavendish*) seedlings. The result was confirmed by real-time PCR analysis, in addition to tissue specific expression, expression differences among *MaMRS2* members were also observed under Mg deficiency conditions. These results showed that Mg transporters may play a versatile role in banana growth and development, and our work will shed light on the functional analysis of Mg transporters in banana.

## Introduction

Magnesium (Mg) is essential in plant growth and development, and cannot be substituted [1]. The major function of Mg in green leaves is to form the central atom of chlorophyll. Mg is also involved in protein and nucleic acid synthesis, and it acts as a bridging element for the aggregation of ribosome subunits [2] and a conformation stabilizer [3] respectively. Furthermore, Mg directly participates or increases the reaction rate of key enzymes, such as isocitrate lyase and ribulose bisphosphate carboxylase [4].

A substantial proportion of Mg in a plant cell is involved in the regulation of its cellular pH and cation–anion balance [5]. The concentration of Mg in metabolic pools, such as the

**Funding:** This work was supported by the National Natural Science Foundation of China (grant numbers 31971520 and 31460089). This work was supported by the National Natural Science Foundation of China (grant numbers 31760549) The funders had no role in study design, data collection and analysis, decision to publish, or preparation of the manuscript.

**Competing interests:** The authors have declared that no competing interests exist.

cytoplasm and chloroplast, are strictly regulated. And magnesium transporter (MGT) plays a vital role in maintaining the equilibrium and homeostasis of Mg in plants [6, 7].

Mg transporters exist in almost all organisms. So far, four kinds of MGTs—*CorA*, *MgtA*, *MgtB*, and *MgtE*—have been found in bacteria [8, 9]. Among them, expression of *MgtA*, *MgtB*, and *MgtE* was induced by Mg-deficient conditions and *CorA* served as the main MGT under normal conditions [6]. The CorA family members contain a 2-TM-GxN domain, and their functional membrane channels are formed by oligomers containing four or five subunits [10, 11]. In yeast, the basic system to maintain Mg cellular homeostasis consists of five MGTs belonging to the CorA superfamily. Among them, the *ALR* genes (*ALR1* and *ALR2*) are located in the plasma membrane where they mediate the uptake of various divalent cations including Mg [12, 13]. Both *MRS2* and *LPE10* are located in mitochondrial intima wherein each plays an indispensable role in Mg homeostasis in yeast cells [14]. *MNR2* is located in the vacuole membrane, where it regulates the storage of Mg in yeast cells [15].

In plants, extensive study of MGTs has been conducted in Arabidopsis (*Arabidopsis thaliana*), rice (*Oryza sativa*), and maize (*Zea mays*) [16–19]. MGTs in sugarcane (*Saccharum spontaneum*), pear (*Pyrus bretschneideri*), tomato (*Solanum lycopersicum*), and other plant species [20–22] have also been characterized recently. A total of 11 *MGT* genes have been found in Arabidopsis, including nine MGT proteins and two hypothetical ones [16, 17]. In other plants such as rice, maize, pear, and sugarcane, 9, 12, 16, and 10 MGTs were found respectively [18–21]. Two transmembrane (TM) domains were present in the plant MGTs studied to date. At the same time, GMN (glycine–methionine–asparagine) domain was also present in most of the identified MGT members of plants [23], although slight variations existed in some members in rice [18] and maize [19]. Beside sequence difference, diverse expression patterns of *MGT* genes have been detected in plants. For example, in Arabidopsis, *AtMGT7b* expressed only in roots and flowers [24], *AtMGT5/AtMRS2-6* expressed in pollen in the early stages of flower development, while *AtMGT9/AtMRS2-2* abundantly expressed in the tapetum. Similarly, rice *OsMRS2-5* is expressed in tissues other than flag leaves, and *OsMRS2-8* is expressed in all tissues but rarely in leaves [18], and pear *PbrMGT4* is expressed only in leaves while *PbrMGT16* is expressed solely in pistils [21]. In addition, the expression of MGTs in plants is also affected by circadian rhythm and light [18, 20, 25]. The versatile Mg transporter genes with diverse expression pattern in plants contribute to their functional complexity. And some genes even participate in pollen development [26, 27].

Banana (*Musa acuminata*) is an important economic crop in tropical regions, where soil acidification is a serious problem [28]. The field of many banana plantations is acidic with a low cation exchange capacity; hence, Mg is highly prone to leaching from soil, especially after heavy rainfall events in the rainy season [29]. Imbalanced fertilization practices aggravates the likelihood of Mg deficiency, as a result, Mg deficiency is now a major contributor to banana yield reduction [30]. Therefore, improving the efficiency of Mg utilization in banana has immediate practical significance. Clearly, MGT plays a vital role in Mg nutrition maintenance in plants. Although the whole-genome sequencing of banana cultivars has been completed [31], the MGT in banana cultivars has not been systematically studied, and the function of different MGT members is not yet clear. The objectives of this study were thus: 1) To isolate the putative members of the banana *MaMRS2* genes, and analyze their physical and chemical properties and functional structures via bioinformatics methods; 2) To clone the genes and determine their respective expression patterns; 3) To identify the function of the genes. This study could provide timely clues to better understand the role of different MGTs in banana.

**Table 1. List of *MaMRS2* genes in the banana genome.**

| Name[a] | Locus tag | Names in Banana Hub | TM domains | pI | Mm (kDa) | Gene length (bp) | Protein length(aa) | Chr[b] | Location | Putative subcellular localization[c] |
|---|---|---|---|---|---|---|---|---|---|---|
| MaMRS2-1 | LOC104000091 | GSMUA_Achr10P05520_001 | 2 | 4.72 | 48.1 | 5914 | 427 | 10 | 15317184~15322824 | P |
| MaMRS2-2 | LOC103972475 | GSMUA_Achr11P26270_001 | 2 | 4.53 | 49.1 | 6412 | 444 | 11 | 25114115~25121080 | P |
| MaMRS2-3 | LOC103983579 | GSMUA_Achr5P01890_001 | 2 | 4.92 | 54.8 | 4877 | 495 | 5 | 1156746~1163541 | Ch |
| MaMRS2-4 | LOC103982423 | GSMUA_Achr4P29320_001 | 2 | 4.49 | 42.6 | 5878 | 379 | 4 | 27328932~27334470 | P |
| MaMRS2-5 | LOC103970801 | GSMUA_Achr11P09590_001 | 2 | 4.81 | 46.5 | 3317 | 414 | 11 | 7479204~7483120 | P |
| MaMRS2-6 | LOC103993960 | GSMUA_Achr8P03230_001 | 2 | 9.16 | 54.6 | 11794 | 495 | 8 | 2258592~2271181 | P |
| MaMRS2-7 | LOC103989965 | GSMUA_Achr1P23260_001 | 2 | 4.87 | 49.6 | 8240 | 443 | 1 | 17454984~17462546 | Cy |
| MaMRS2-8 | LOC103986367 | GSMUA_Achr5P26100_001 | 2 | 4.51 | 48.6 | 5909 | 439 | 5 | 27012863~27018813 | P |
| MaMRS2-9 | LOC103982990 | GSMUA_Achr4P22510_001 | 2 | 5.21 | 47.6 | 8016 | 421 | 4 | 22793450~22801155 | Ch |
| MaMRS2-10 | LOC103997852 | GSMUA_Achr9P13370_001 | 2 | 4.82 | 47.8 | 4812 | 427 | 9 | 8683743~8688133 | P |

[a] Names assigned to banana *MRS2* genes in this study.

[b] Chromosomal localization of banana *MRS2* genes.

[c] WoLF PSORT predictions: P (plasma membrane), Ch (chloroplast), Cy (Cytoskeleton).

## Results

### Identification of *MRS2* family genes in banana

Ten putative Mg transporter sequences from banana's genome were obtained using the CorA/MGT/MRS2 sequences of Arabidopsis, rice, maize, and yeast as queries. These genes were named according to the nomenclature of Arabidopsis and rice (Table 1). The isoelectric point, the amino acid length and the molecular weight of the predicted proteins ranged from 4.51 to 9.16 (pI), from 379 to 495 (aa), and from 42.57 to 54.83 (kDa), respectively. Subcellular localization of 10 members predicted that seven proteins located on the plasma membrane, while MaMRS2-7 located in the cytoskeleton, MaMRS2-3 and MaMRS2-9 resided on the chloroplast. TM domain analysis of MaMRS2 indicated that all 10 members contain two hypothetical TM domains in the C-terminal (S1 Fig). Sequence alignment of the MaMRS2 proteins showed they had high sequence similarity, with all members having a GMN functional domain except MaMRS2-4, which instead had a mutated motif AMN (alanine–methionine–asparagine) (Fig 1).

### Comparative analysis of the *MRS2/MGT* genes from Arabidopsis, rice, maize, yeast, and banana

Phylogenetic trees were constructed using MGTs of banana, Arabidopsis, rice, maize and yeast. Among them, MGTs of yeast was selected as the outgroup (Fig 2). The plant MGT family proteins were divided into five branches, each branch contained a different number of MGT members derived from different plant species. For each species, just one MGT family member

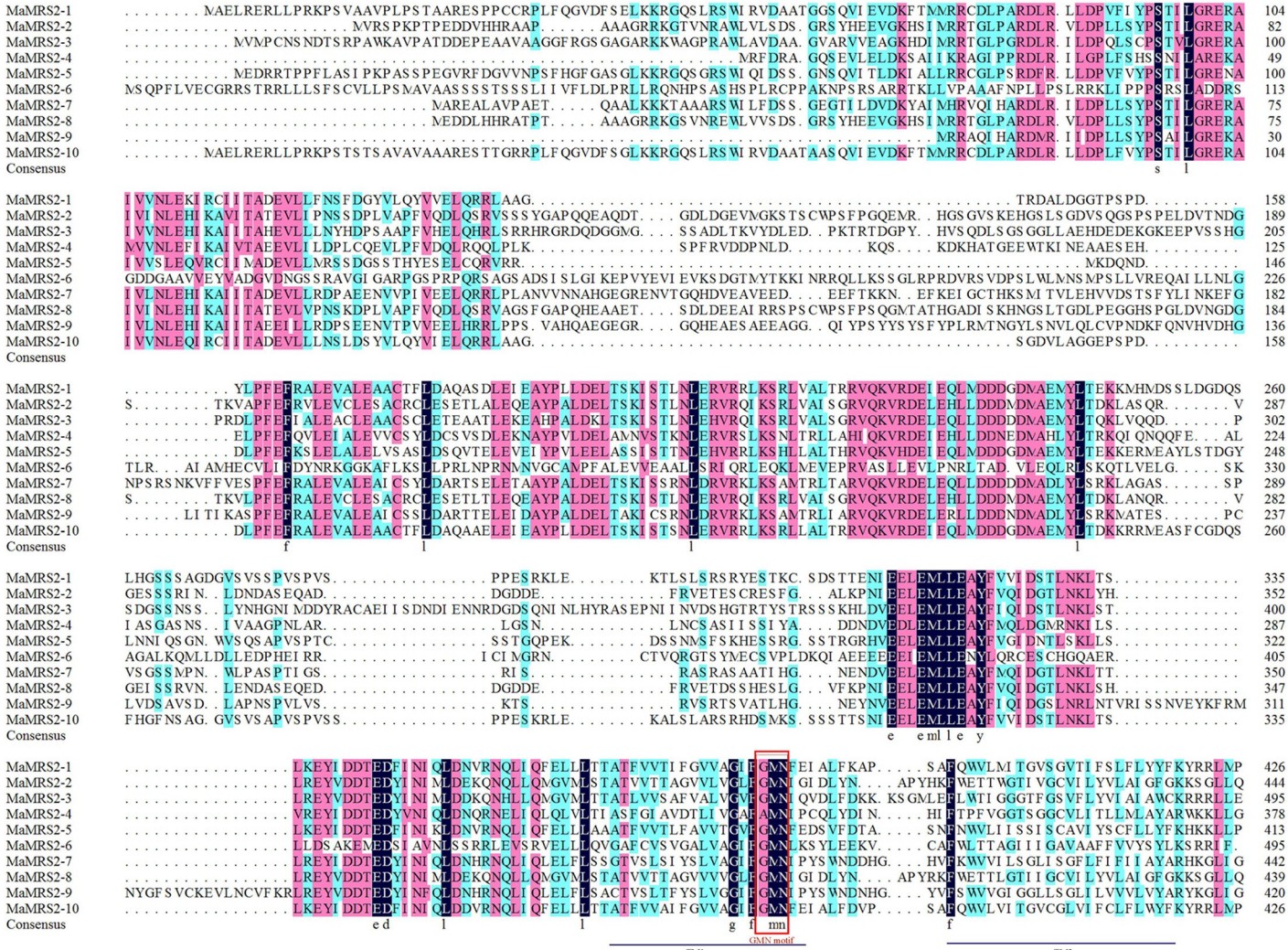

**Fig 1. Multiple sequence alignments of MaMSR2 proteins.** Alignment was performed using DNAMAN software. The identical, conserved and less conserved amino acid residues are indicated by dark, cherry red and cyan background colors, respectively. The conservative GMN motif was indicated and the TM domains are underlined.

was respectively distributed in branch C, which diverged from other MGTs, possibly indicating a unique role of this member protein. The most abundant Mg transporters of banana were distributed in the G and H branches, each having three members. In each group, the similarity of sequences from the same plant species was higher than that from other plant species.

## Phylogenetic relationships, gene structure, and motif analysis of MaMRS2 family members

The phylogenetic tree (Fig 3A) and genetic structural analysis of 10 MaMRS2 protein members (Fig 3B) in banana showed that proteins from the same branch had similar structure, but this differed greatly when members from different branches were compared. For example, MaMRS2-10, MaMRS2-1, and MaMRS2-5 were on the same branch, whose number and distribution of exons were similar; conversely, MaMRS2-6 and MaMRS2-4, which were on two different branches, varied greatly in the number of exons and their position. To further

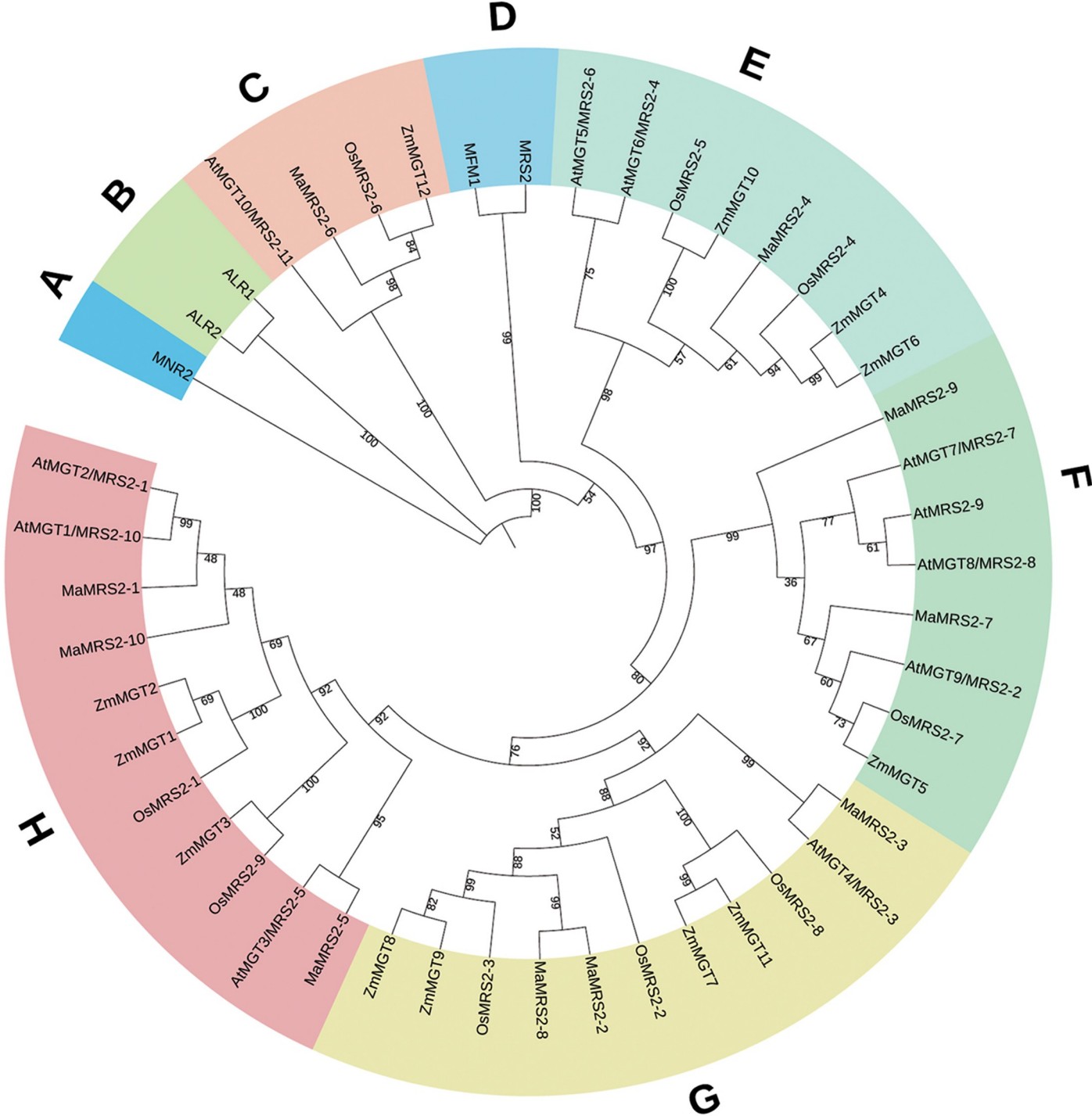

**Fig 2. Phylogenetic analysis of Arabidopsis, rice, maize, yeast and banana CorA/MRS2/MGT members.** The Neighbor-Joining tree, which includes 10 MaMRS2 protein from banana, 11 MRS2/MGT proteins from Arabidopsis, 9 MRS2/MGT proteins from rice, 12 MRS2/MGT proteins from maize and 5 MRS2/MGT proteins from yeast, was constructed using MEGA X. A, B, C, D, E, F, G and H represent the different clades of the MRS2/MGT family in these five species.

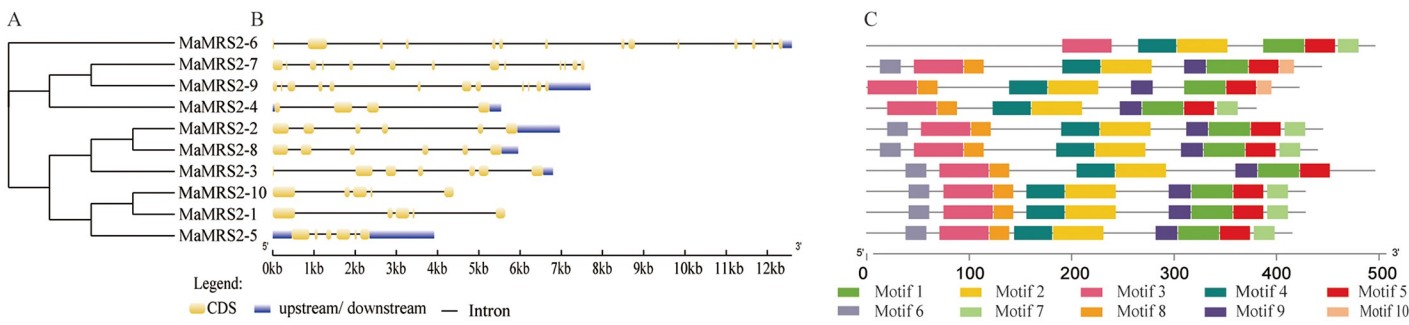

**Fig 3. Phylogenetic relationships (A), gene structure (B) and motif analysis (C) of MaMRS2 family members.**

understand the relationship between members of the *MaMRS2* gene family, motif analysis was performed for each MaMRS2 member (Fig 3C). These results indicated that 10 putative motifs were distributed in the MaMRS2 proteins. Common motifs such as motif 1/2/3/4/5 existed in all Mg transporter members. In particular, motif1 and motif5 near the 3' end contained the CorA/MRS2 functional structure. At the same time, different members contained different motifs. For example, motif 6 was absent from MaMRS2-6 and MaMRS2-9, and motif 10 was found only in MaMRS2-7 and MaMRS2-9. Together, these results clearly indicated the great versatility characterizing banana's Mg transporters.

The Phylogenetic tree was constructed using the Neighbor-Joining method with 1000 bootstrap replicates in the MEGA X software. Then the gene structure was performed using GSDS program. MEME program and TBTools were used to illustrate the motif analysis results. Yellow boxes and black lines represent exons and introns, respectively, blue boxes indicate 5' or 3' untranslated regions (UTRs) and different motif was painted with different color.

## Chromosomal location and gene duplication

The position information of a given *MaMRS2* gene on a chromosome was obtained from NCBI. The ten *MaMRS2* genes were distributed among seven chromosomes of banana: two *MaMRS2* genes on the chromosome 4, 5, and 11, while chromosome 1, 8, 9, and 10 each contained a single *MaMRS2* gene. To understand the expansion mechanism of these *MaMRS2* genes, the collinear relationship between them was analyzed via MCScanX software (Fig 4). These results revealed gene duplication events, in that a total of 12 pairs of duplication relationships between *MaMRS2* genes occurred on six chromosomes. No tandem duplication relationships were found, nor was there evidence for duplication between *MaMRS2* members on the same chromosome. Collectively, these results indicated that some *MaMRS2* genes have duplicated more than once, for which multiple rounds of whole genome duplication events might be responsible. The Ka/Ks of five homologous Mg transporters was less than 0.3 (S1 Table); however, the Ka/Ks ratio of the others was not obtained from the data because the same sense mutation could not be detected, indicating that the *MaMRS2* gene evolved mainly under purify selection.

## Prediction of cis-acting elements in the promoter region of *MaMRS2* genes

To understand the transcriptional regulation of the *MaMRS2* family genes in banana, about 2-kb upstream promoter region of *MGT* genes was selected and submitted to the PlantCARE website for analysis, and their functional regions were visualized with TBTools (Fig 5). The promoter region of the *MaMRS2* genes contained cis-acting elements of light response,

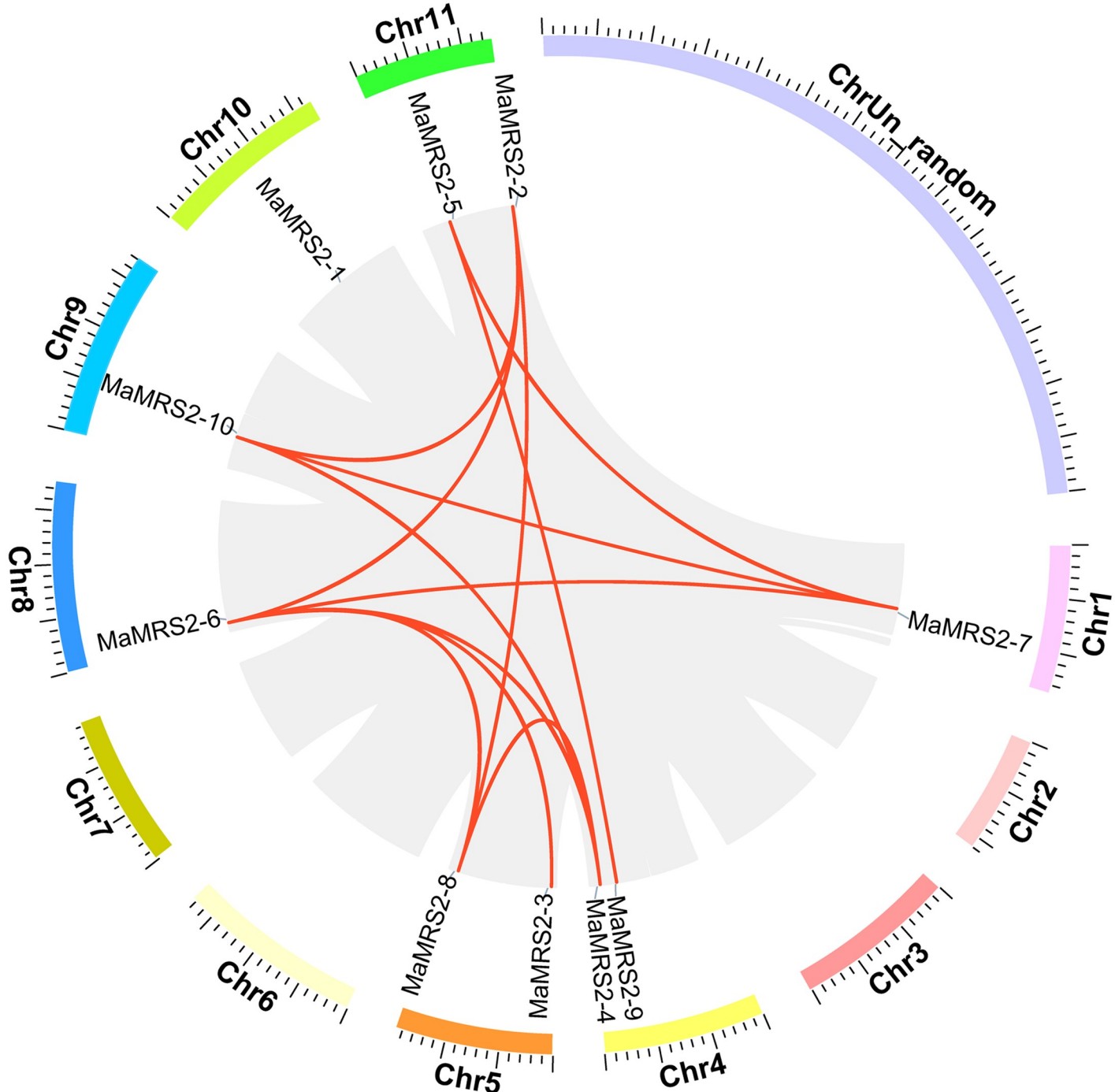

**Fig 4. Chromosomal location and gene duplication of *MaMRS2* genes in the banana genome.** The *MaMRS2* gene is located on different chromosomes. The number of chromosomes is shown on the outside, and the different colors represent different chromosomes. The grey region is the collinearity of the banana genome, highlighting the collinearity between the *MaMRS2* genes was highlighted with red lines.

circadian rhythm, hypothermia, defense and stress responses, and phytohormones, among others. Some cis-acting elements such as light response elements were common to the promoter regions of all *MaMRS2* gene members, whereas others were restricted to a certain *MaMRS2* member. For example, the circadian rhythms response element is found only in the

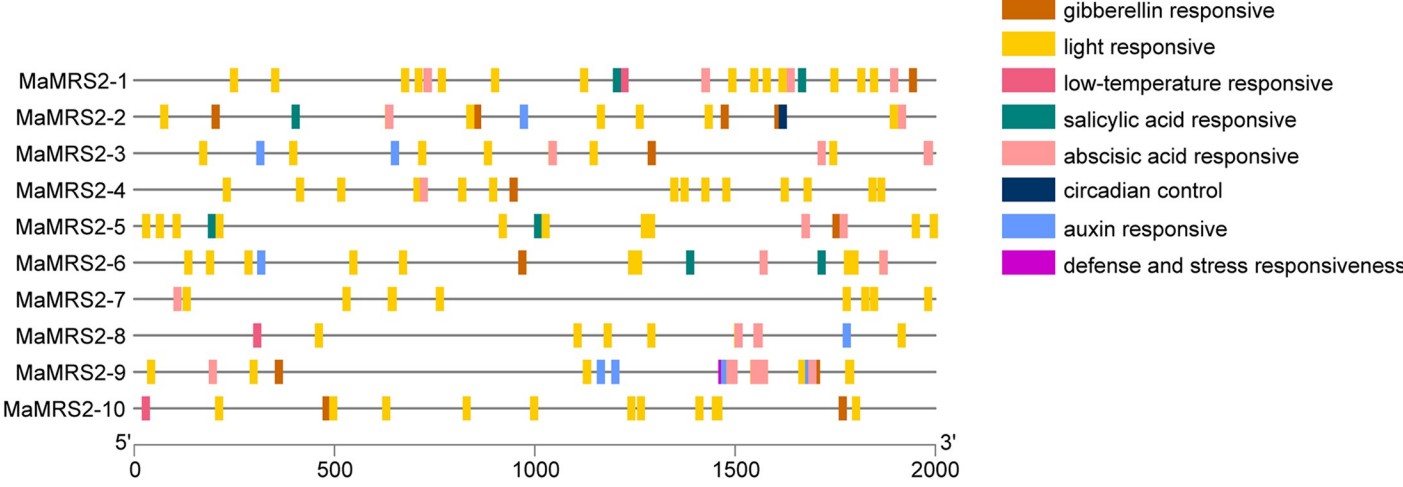

**Fig 5. Analysis of cis-acting elements of members of the *MaMRS2* gene family.** Different colors represent cis-acting elements of different functions.

promoter region of *MaMRS2-2*, and the defense and stress response element is only found in the promoter region of *MaMRS2-9*. These results suggested that, even though Mg transport is a light-regulated process, some *MaMRS2* genes could also participate in other biological processes involving Mg during banana growth and development.

## Function complementation of *MaMRS2* genes

Based on phylogenetic analysis (Fig 2), three *MaMRS2* genes (*MaMRS2-1*, *MaMRS2-4*, *MaMRS2-7*) in three different branches (H, E, F) were cloned and sequenced (S2 Fig, S1 Test). Their coding regions were amplified and subcloned into pTrc99A for functional analysis by using the *Salmonella typhimurium* mutant MM281. The positive control *S. typhimurium* (wild type MM1927) grew well on medium with 0.01 mM of Mg. By contrast, negative controls— MM281, and MM281 transformed with empty pTrc99A—could hardly grow on medium with Mg less than 10 mM. Three *MaMRS2* genes (*MaMRS2-1*, *MaMRS2-4*, *MaMRS2-7*) were transferred into MM281 respectively, and their growth was restored on the medium containing low Mg levels, indicating these *MaMRS2* members encoded Mg transporters. Nevertheless, the Mg transfer efficiency of different members varied greatly, as evidenced by the different growth rate of the recovery strains under the same Mg content, both on agar plate and in liquid culture (Fig 6A and 6B).

## *MaMRS2* gene expression in different tissues of banana

To further elucidate the function of *MaMRS2* genes in banana, their expression in different tissues of banana cultivar 'Baxijiao' (*Musa spp. AAA Cavendish*) seedlings were measured via semi-quantitative PCR respectively, in which *MaTUB* served as the reference gene. The results showed that all *MaMRS2* genes could be expressed, but their expression patterns differed markedly. For instance, *MaMRS2-2* expression was only detected in corm with low abundance. In stark contrast, *MaMRS2-1*, *MaMRS2-4*, *MaMRS2-8*, and *MaMRS2-9* were constitutively expressed, although their expression differed slightly among the root, pseudo stem, corm, and leaves (Fig 7).

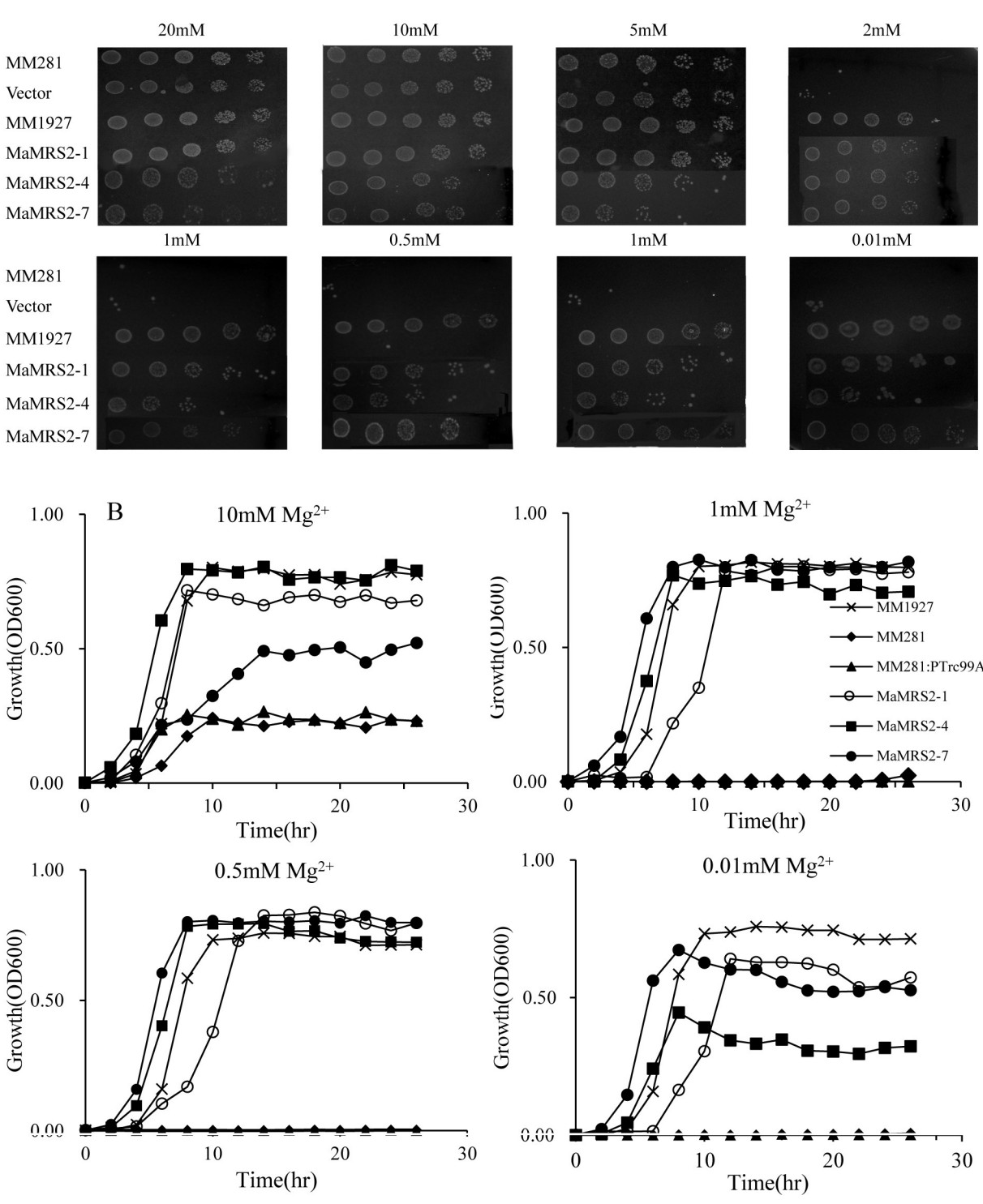

**Fig 6. Complementary analysis of the *MaMRS2* genes.** MM1927 was the wild type and used as a positive control, MM281 and MM281 transformed with an empty pTrc99A were used as negative controls. (A) Functional verification on N minimal solid medium containing 20, 10, 5, 2, 1, 0.5, 0.1, 0.01 mM MgSO4. The bacterial was diluted sequentially 10-fold from left to right. (B) Growth curves were performed in N-minimal liquid medium containing 10, 1, 0.5 and 0.01mM MgSO4, and the cell density was monitored at OD600 every 2 hours. Data was an average of three independent experiments, and the different icons in the figure represent the average of three repetitions.

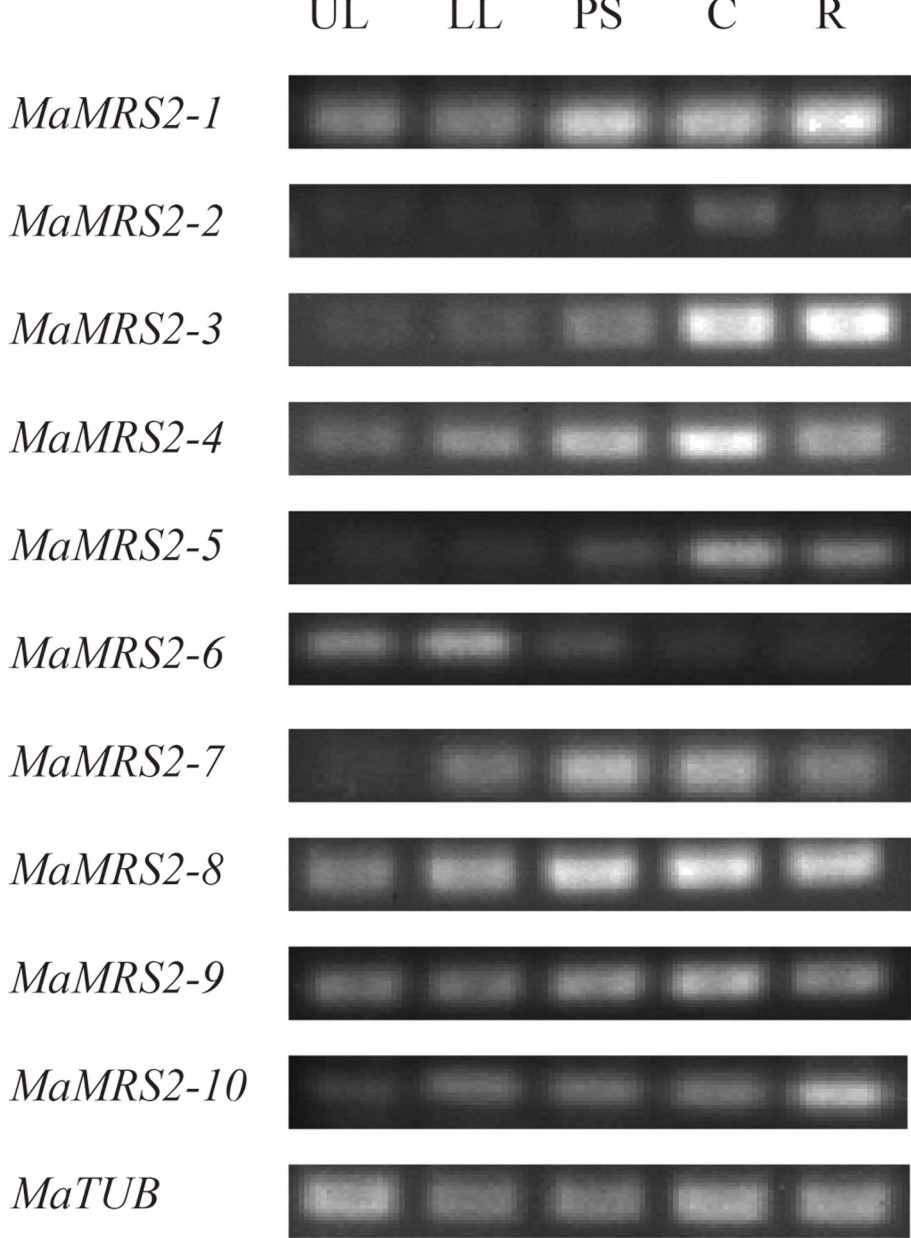

**Fig 7. Expression of *MaMRS2* gene in different tissues of Baxijiao seedlings.** UL, LL, PS, C and R represent upper leaf, lower leaf, pseudo stem, corm and root respectively.

## *MaMRS2* gene expression under Mg deficiency

Gene expression analysis was also carried out via qRT-PCR, in five different tissues (upper leaf, lower leaf, pseudo stem, corm, root) from banana seedlings under normal or Mg-deficient conditions. According to the results, Mg deficiency greatly affected the expression of *MaMRS2* genes in the upper leaf, lower leaf, pseudo stem, corm, and root of banana (Fig 8). Compared with their controls, when deprived of Mg, *MaMRS2-1* and *MaMRS2-10* underwent down-regulated expression in all tissues tested, whereas *MaMRS2-5* and *MaMRS2-7* were up-regulated in leaves, pseudo stem, and corm parts. The contrasting expression pattern of different Mg

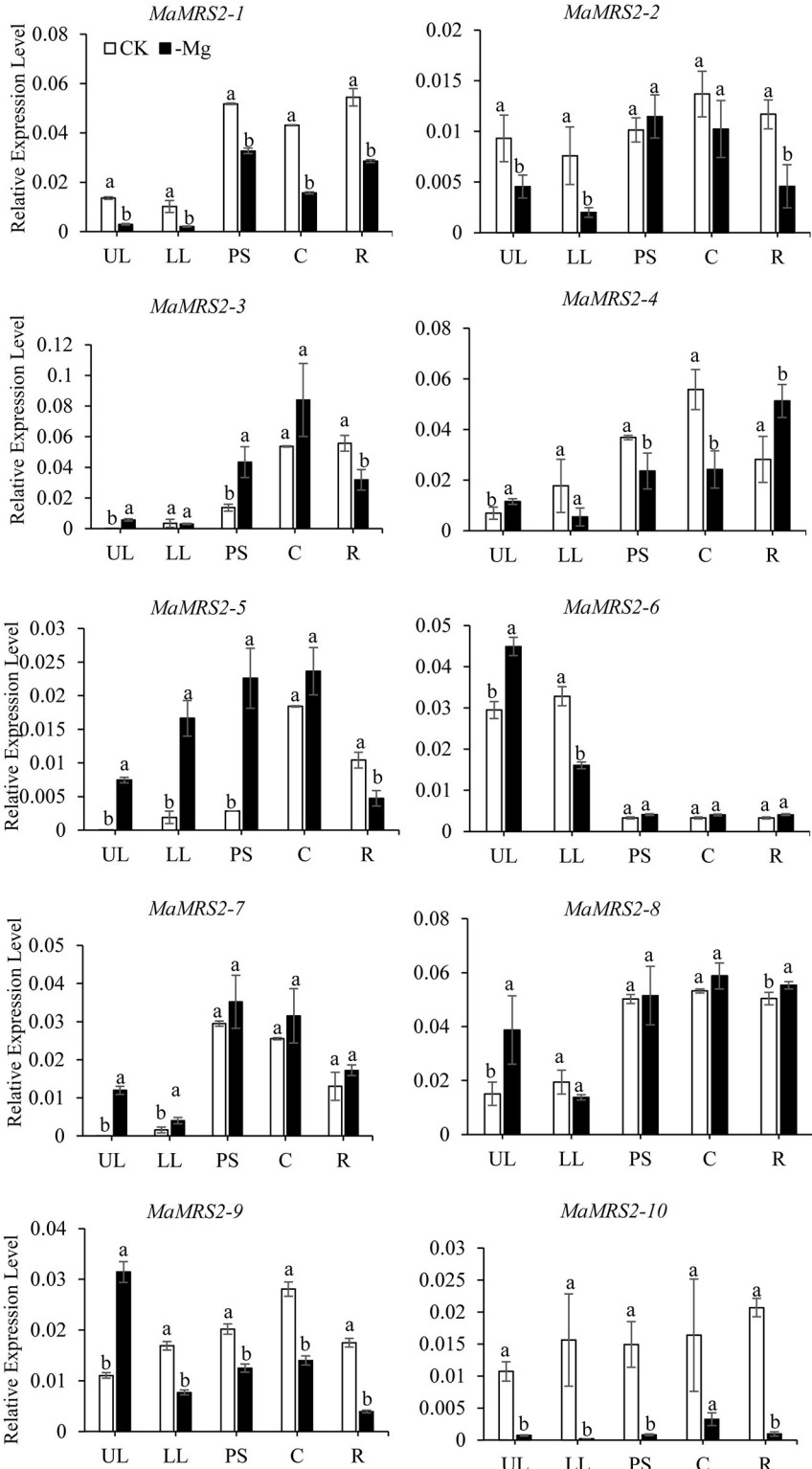

**Fig 8. Relative expression level of *MaMRS2* gene in different tissues of Baxijiao seedlings under magnesium deficiency.** The CK indicated the value under normal growth conditions as a control, and the -Mg indicated the relative expression level under the complete absence of magnesium ion condition. UL, LL, PS, C and R represent upper leaf, lower leaf, pseudo stem, corm and root respectively.

transporter genes among differing tissues under Mg deficiency (Fig 8) indicated these genes is probably not only involved in Mg uptake and transport, but also participated in Mg allocation among different tissues.

## Discussion

Mg is the essential nutrient element for plant growth and development and Mg transporter genes played an indispensable role in Mg absorption and transport. In model plants such as Arabidopsis, rice, and maize, Mg transporter genes have been extensive studies [16–19]. Recently, they were also functionally analyzed in tomatoes [22], pears [21], and sugarcane [20]. However, little is known about the genes in banana, a tropical crop with high biomass, which is vulnerable to Mg deficiency in the field. In our study, 10 *MaMRS2* genes in banana genome were identified based on their sequence similarity, and they could be divided into five evolutionary branches, which was not beyond the range of the phylogenetic relationship using other plant species, such as Arabidopsis and rice [18, 19, 22]. Accordingly, all MaMRS2 members had a two transmembrane domain and a highly conserved GMN domain. The only exception is that MaMRS2-4 had a mutated AMN domain. This phenomenon was also observed in the ZmMGT6 of maize [19] and in OsMRS2-4 and OsMRS2-5 of rice [18]. Genomic collinearity showed that duplication relationships occurred between MaMRS2 family members across chromosomes, but no signs of a tandem duplication relationship (Fig 4), a pattern generally exists among other gene family members of banana [32]. The sequence similarity and phylogenetic relationship showed that all Mg transporters analyzed in this study have the same ancestor, and it can be deduced that the main function of these genes is almost the same.

To functionally analyzed the MGTs, both yeast and *S. typhimurium* complementary experiments can be used although slight differences were observed using these two systems [18, 19]. In maize and Arabidopsis, five MGT members were identified using *S. typhimurium* complementary experiments respectively [17, 19, 24, 26, 27, 33]. And a total of 9 Arabidopsis and 4 rice *MGTs* were identified via yeast complementary experiment. In our experiment, three of 10 *MaMRS2* genes in banana (*MaMRS2-1*, *MaMRS2-4* and *MaMRS2-7*) located in different clades were selected, which functionally complemented the Mg transporter genes in *S. typhimurium* respectively. The affinity of these proteins to Mg was not the same, *MaMRS2-1* and MaMRS2-7 have a higher affinity for Mg than MaMRS2-4 (Fig 6). Similar results were obtained in maize in that *ZmMGT1* (homologous to *MaMRS2-1* in banana) and *ZmMGT5* (homologous to *MaMRS2-7*) had stronger complementary effect than *ZmMGT6* (homologous to *MaMRS2-4*) [19]. These results indicated that the affinity or efficiency of different MGTs during Mg transport, absorption, and storage differed greatly, and each *MaMRS2* genes may have a distinctive role. The complementary lines incorporated with *MaMRS2-4* and *MaMRS2-7* could grow well under low Mg concentration conditions, but they had lower growth rate when compared with the mutant control under solid growth medium with higher Mg concentration (10 mM and 20 mM) (Fig 6A). These results indicated that both *MaMRS2-4* and *MaMRS2-7* could transport Mg, at the same time, the exogenous Mg transporters from banana might play its role in excessive Mg accumulation and lead to the toxic effect in bacteria under high Mg conditions.

The detailed analysis indicated that although the overall functional similarity presented in MGT genes, the transcriptional changes and the detailed function varied greatly. Some genes seem to play a universal role. For example, *MaMRS2-6* was highly expressed in leaves of banana seedlings (Fig 7), and its counterpart, which was deemed to be involved in photosynthesis in maize, Arabidopsis and rice, was also highly expressed in leaves [18, 25, 34]. Similarly, banana *MaMRS2-4* gene was significantly up-regulated in the banana roots under

Mg-deficiency (Fig 7), *MaMRS2-4* is homologous to *AtMRS2-4/AtMGT6* in Arabidopsis, *ZmMGT10* in maize and *SlMGT4-1* in tomato, whose respective up-regulation was observed in the roots when deprived of Mg [22, 33, 35, 36]. However, some homologous genes diverged during their evolution probably because of the plant specific requirement. *MaMRS2-2* was specifically expressed in the corm of banana plants under normal Mg supply conditions, and this expression pattern was negligibly changed by the Mg deficiency treatment (Fig 8). While its homologous rice *OSMGT1/OsMRS2-2* gene, which linked with the response to salt and aluminum stress [37], was highly expressed in rice roots and leaves. The homologous tomato gene, *SlMGT4-1*, expressed mainly in roots and shoots under Mg deficiency conditions [22]. The expression difference may be arise from cis-acting elements, for a large variety of regulatory elements, were presented in the promoter region of banana's *MaMRS2* genes (Fig 5). From the above data, we can see that *MaMRS2* is similar to the MGT protein of other plant species not only in terms of its sequence structure but also in its expression pattern and function. However, the specific functioning of *MaMRS2* in banana's growth and metabolism and how it affects this plant's transport and absorption of Mg await further investigation and experimental verification.

Field practice indicates that Mg-deficiency is one of the most common problems during banana' growth and development [38]. How to improve their Mg nutrient utilization efficiency is thus of vital importance. Despite field or foliar applications of Mg fertilizer being effective means to enhance Mg nutrient status [39], genetic improvement is considered the most cost-effective way, for which MGT is a potential candidate for breeding and engineering. The identification and functional analysis of Mg transporters in this work provides not only clues to Mg absorption and transport mechanisms in banana but also a larger pool for candidate gene selection by crop researchers.

## Conclusion

In summary, the MaMRS2 protein family has 10 members with five clades. Each member has two TM domains and a GMN domain, except *MaMRS2-4*. *MaMRS2* genes are unevenly located on six chromosomes, and there are multiple cis-acting elements in their promoters. Three *MaMRS2* gene members (*MaMRS2-1*, *MaMRS2-4*, *MaMRS2-7*) in the three branches (H, E, F) indicated that they are indeed Mg transporters based on functional complementation analysis. The expression *of MaMRS2* genes varied greatly in different tissue or under Mg deficiency conditions. Therefore, *MaMRS2* genes participate in Mg absorption and transport in banana and each probably played a distinct role during plant growth and development.

## Materials and methods

### Sequence origin and bioinformatics analysis

The whole genome sequence of dwarf banana (*Musa acuminate*) and the CorA/MGT/MRS2 protein sequence of yeast were obtained from NCBI (https://www.ncbi.nlm.nih.gov/). The MGT protein sequences of *Arabidopsis thaliana*, rice (*Oryza sativa*) and maize (*Zea mays*) were downloaded from the Phytozome (https://phytozome.jgi.doe.gov/pz/portal.html) website. The TBLASTN program, in BioEdit v.7.0.4 software, was used under Windows operation system to obtain candidate Mg transporter protein sequences in banana, for which e-values were set to less than –10 [40]. The redundant sequences were manually removed; the remaining sequence of each candidate member was further queried in the Pfam (http://pfam.xfam.org/) [41], SMART (http://smart.embl-heidelberg.de/) [42], and the Conserved Domains Servers (https://www.ncbi.nlm.nih.gov/Structure/cdd/wrpsb.cgi) to find the corresponding CorA/MRS2 domain. Those proteins lacking this functional domain were deleted. Then, the TM of

candidate MaMRS2 members was predicted by the TMHMM Server v.2.0 online program (http://www.cbs.dtu.dk/services/TMHMM/) [43]. The isoelectric point and molecular mass (kDa) of each candidate protein was calculated with BioXM v.2.6 software [44]. The subcellular localization of *MaMRS2* members was predicted by the WolF PSORT program (https://wolfpsort.hgc.jp/) [45].

The sequences of MaMRS2 proteins were compared by multiple sequence alignment with DNAMAN v.2.6 [46]. A phylogenetic tree was built using the neighbor-joining method in the MEGA X program [47–49]. Its bootstrap value was set to 1000, and the output phylogenetic tree file was graphed by the online iTOL program (https://itol.embl.de/) [50].

The downloaded CDs' sequences and gene sequences, as well as the phylogenetic tree generated by MEGA X software, were then submitted to GSDS2.0 (http://gsds.cbi.pku.edu.cn/) [51] for gene structure analysis. The MaMRS2 protein sequences were submitted to the MEME v.5.1.0 (http://meme-suite.org/tools/meme) [52] to detect their motifs, carried out with default parameters.

To analyze the whole-genome duplication of the banana, MCScanX software in the Linux operating system was used [53]. Collinearity and location mapping of *MaMRS2* genes and their corresponding Ka/Ks calculations were performed using TBTools software [54].

The upstream 2-kb sequence of each *MaMRS2* gene was downloaded, and then analyzed with the PlantCARE (Plant Cis Acting Regulatory Element) server (bioinformatics.psb.ugent. be/webtools/plantcare/html/) to predict its cis-acting element. The cis-acting components of interest were visualized using TBTools [54].

## Sample preparation and gene expression analysis

Banana cultivar 'Baxijiao' (*Musa spp. AAA Cavendish*) seedlings were sand-cultured in the greenhouse at the Hainan University from July 7th to September 29th, 2018. We use Hoagland's nutrient solution to treat the control, which contained 6 mM $KNO_3$, 4 mM Ca $(NO_3)_2 \cdot 4H_2O$, 2 mM $NH_4H_2PO_4$, 4 mM KCl, 1 mM $MgSO_4 \cdot 7H_2O$, 60 μM Fe-EDTA, 25 μM $H_3BO_3$, 2 μM $MnSO_4 \cdot H_2O$, 2 μM $ZnSO_4 \cdot 7H_2O$, 0.5 μM $CuSO_4 \cdot 5H_2O$, and 0.05 μM $H_2MoO_4$. For Mg deficiency treatments, 1 mM $MgSO_4 \cdot 7H_2O$ in nutrient solution was replaced by 1 mM $K_2SO_4$, and 4 mM $KNO_3$ and 1 mM $NH_4NO_3$ were used to maintain the same N supply level. Each time of fertilizer application, 500 mL nutrient solution was supplied to each pot. Every seventh day each pot was flushed with deionized water to wash out accumulated nutrients in quartz sand. After 12 weeks of plant growth, the roots, pseudo stem, corms, upper leaves (the second fully-expanded blade from the top), and lower leaves (the sixth fully-expanded blade from the top) were collected, immediately frozen in liquid nitrogen, and stored at –80˚C until their RNA extractions.

The RNA from each tissue type sample was extracted with the RNAprep Pure Plant Kit (Tiangen), and its reverse transcription conducted using FastKing gDNA-dispelling RT Super-Mix (Tiangen). The primers designed by Primer Premier 5 software [55] can be found in S2 Table. Semi-quantitative RT-PCR was performed to detect the expression patterns of *MaMRS2* genes in different tissues, with a qRT-PCR analysis done to quantify their relative expression levels in five plant tissue types in response to Mg deficiency (0 mM Mg). The qRT-PCRs were all conducted on a PCR automatic serial analysis system LightCycler® 96 (Roche) with TIANGEN SuperReal PreMix Plus (SYBR Green). The expression level of each *MaMRS2* gene was calculated using the $2^{-\Delta Ct}$ method, for which the *TUB* gene served as the internal reference [56]. The bar charts were drawn using Microsoft Excel. Means were compared by the Tukey LSD method, in SPSS v.14.0 software [57] with a significant level set at $P < 0.05$. The data are shown as the mean ± SD of three biological replicates.

### Gene cloning and functional complementation analyses of *MaMRS2* genes

Three representative genes from the H (*MaMRS2-1*), E (*MaMRS2-4*), and F (*MaMRS2-7*) sub-families were amplified, by using a gene-specific primer with endonuclease restriction sites at both ends (S3 Table). Each PCR product was then cloned into the pTrc99A vector. The positive clones of each genes were confirmed by sequencing. The complementation experiments [19] were then carried out to test whether three *MaMRS2* gene members were able to transport Mg. *Salmonella typhimurium* wild-type MM1927 served as the control for the mutant *S. typhimurium* MM281 that had its *CorA*, *MgtA*/*MgtB* gene inactivated (it fails to grow in media with an Mg concentration < 10 mmol/L). The $OD_{600}$ was measured every 2 h to detect the bacteria growth rate in liquid media. The line charts were drawn using Microsoft Excel. The data are shown as the mean of three biological replicates.

## Supporting information

**S1 Fig. Analysis of Transmembrane (TM) domains of the MaMRS2 family proteins using the TMHMM program, red region indicates transmembrane region.**
(TIF)

**S2 Fig. RT-PCR amplification of the entire ORF of three MaMRS2 genes.** The root or leaf cDNAs were used as template. The RT-PCR was performed 35 cycles using gene specific-primer pairs.
(TIF)

**S1 Table. Ka/Ks of MaMRS2 genes.**
(PDF)

**S2 Table. PCR primers used for semi-quantitative RT-PCR and qRT-PCR analysis.**
(PDF)

**S3 Table. PCR primers used for complete CDS amplification.**
(PDF)

**S1 Test. The nucleotide sequences of the three *MaMRS2* genes from sequencing.**
(PDF)

**S1 Raw images.**
(PDF)

## Acknowledgments

The authors would like to thank Professor Legong Li (School of Life Sciences, Capital Normal University of China) for providing the *Salmonella typhimurium* MM281 strain and the plasmid pTrc99A.

## Author Contributions

**Conceptualization:** JiaQuan Huang.

**Data curation:** HaiYan Hu.

**Investigation:** MengYing Tong.

**Methodology:** Wen Liu, YuanHao Ding.

**Project administration:** JiaQuan Huang.

**Resources:** HongSu He, Xinguo Li.

**Supervision:** LiYan Yin.

**Writing – original draft:** MengYing Tong.

**Writing – review & editing:** JiaQuan Huang.

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
