## [Decision Letter · Decision Letter 0]

5 May 2020

PONE-D-20-02641

Identification and functional analysis of the CorA/MGT/MRS2-type magnesium
transporter in banana

PLOS ONE

Dear Dr. Huang,

Thank you for submitting your manuscript to PLOS ONE. After careful consideration, we
feel that it has merit but does not fully meet PLOS ONE’s publication criteria as it
currently stands. Therefore, we invite you to submit a revised version of the
manuscript that addresses the points raised during the review process.

We would appreciate receiving your revised manuscript by Jun 19 2020 11:59PM. When
you are ready to submit your revision, log on to https://www.editorialmanager.com/pone/ and select the 'Submissions
Needing Revision' folder to locate your manuscript file.

If you would like to make changes to your financial disclosure, please include your
updated statement in your cover letter.

To enhance the reproducibility of your results, we recommend that if applicable you
deposit your laboratory protocols in protocols.io, where a protocol can be assigned
its own identifier (DOI) such that it can be cited independently in the future. For
instructions see: http://journals.plos.org/plosone/s/submission-guidelines#loc-laboratory-protocols

We look forward to receiving your revised manuscript.

Kind regards,

Anil Kumar Singh, Ph.D.

Academic Editor

PLOS ONE

Journal Requirements:

4. Please ensure that you refer to Figure 1 in your text as, if accepted, production
will need this reference to link the reader to the figure.

Reviewers' comments:

Reviewer's Responses to Questions

**Comments to the Author**

1. Is the manuscript technically sound, and do the data support the conclusions?

Reviewer #1: No

Reviewer #2: Yes

2. Has the statistical analysis been performed
appropriately and rigorously? 

Reviewer #1: No

Reviewer #2: No

3. Have the authors made all data underlying the
findings in their manuscript fully available?

Reviewer #1: No

Reviewer #2: Yes

4. Is the manuscript presented in an intelligible
fashion and written in standard English?

Reviewer #1: No

Reviewer #2: No

5. Review Comments to the Author

Reviewer #1: 1. The author said that these genes were not only involved in Mg uptake
and transport, but also participated in Mg allocation among banana’s root, pseudo
stem, corm, and leaf components，which required more biological experiments to
support it

2.need adding bilogical stastics for q-PCR data

3. DISCUSSION should be rewritten.

4. writting and and orgnization should be further improved.

Reviewer #2: The authors characterized the Magnesium transporter MaMRS2 protein
family. 10 MaMRS2 genes in banana (Musa acuminata) were identified. The
physicochemical properties, location on chromosomes, gene structure, cisacting
elements, and replication relationships between these ten members were analyzed. The
tissue-specific expression pattern was analyzed. Three genes MaMRS2-1, MaMRS2-4, and
MaMRS2-7 were cloned and complemented with triple MGT mutant of Salmonella
typhimurium. This study could help in understanding the function of each MaMRS2 gene
in development or stress conditions.

The manuscript is poorly written and needs to be largely reworked.

1. Rewrite the line: These results should be helpful to further research work on Mg
transporters in banana crops.

2. The introduction is very messy, especially the expression summary of MGTs in
different plants. Rewrite the introduction.

3. Units are missing: The isoelectric point of the predicted proteins and their
corresponding amino acid length ranged respectively from 4.51 to 9.16 (pI) and from
379 to 495 (aa).

4. Rewrite: Figure 2 shows the phylogenetic tree built for Mg transporter protein
sequences from banana, Arabidopsis, rice, maize, and yeast, with yeast Mg
transporters as the outgroup.

5. Chromosomal location and gene duplication: Authors mentioned gene duplication as
gene replication. (5 times). There is a difference between the two terms.

Also, in the discussion part: we found evidence for replication relationships that
occurred between MaMRS2 family members across chromosomes but no signs of a tandem
replication relationship (Fig. 4). This pattern generally exists among other gene
family members of banana [32]. It is reasonable to speculate that multiple rounds of
whole genome replication events have occurred over evolutionary time in banana.

6. Figure legends are not uploaded properly or incomplete:

Fig. 1 Multiple sequence alignments of MaMSR2 proteins. This alignment was performed
using DNAMAN software. The identical, conserved, and less conserved

Fig. 2 Phylogenetic analysis of Arabidopsis, rice, maize, yeast, and banana
CorA/MRS2/MGT members. The neighbor-joining tree, which includes 10 MaMRS2

Fig. 3 Phylogenetic relationships (A), gene structure (B), and motif analysis (C) of
MaMRS2 family members in banana. The phylogenetic tree was constructed using the

Fig. 4 Chromosomal location and gene duplication of MaMRS2 genes in the banana
genome. The MaMRS2 gene is located on multiple chromosomes. The chromosome

Fig. 5 Predicted cis-acting elements in the promoter region of the 10 MaMRS2 genes in
banana. Different colors represent cis-acting elements associated with different

Fig. 6 Complementary analysis of the MaMRS2 genes. MM1927 is the wild type, used here
as a positive control; the MM281 and

Fig. 7 Expression of 10 MaMRS2 genes in different tissues of banana cultivar
‘Baxijiao’ (Musa spp. AAA Cavendish) seedlings. Explain the Abbreviations.

Fig. 8 Relative expression levels of 10 MaMRS2 genes in different tissues of banana
cultivar ‘Baxijiao’ (Musa spp. AAA Cavendish) Explain the Abbreviations.

7. The author mentioned sequences of 7 genes:

Supplementary Test1 The nucleotide sequences of the seven MaMRS2 genes from
sequencing

There are only three sequences >MaMRS2-1, >MaMRS2-4, >MaMRS2-7 as authors
clones only three genes.

8. The authors did complementation assay with MM281 mutant.

Figure 6A: Why the growth of complemented lines is low in higher concentration of
magnesium as compare to Mutant (MaMRS2-4, MaMRS2-7), as Complemented lines are
growing fine in lower concentrations.

Figure 6B: Authors mentioned: The data are shown as the mean ± SD of three biological
replicates,

However, no SD is shown on the line charts.

The growth curve at 0.01 mM Mg2+ shows that complementing lines 2-1 and 2-7 are
growing better than the wild type, which does not correlate with Figure 6A.

9. qRT-PCR analysis of MaMRS2-9 has no error bars; what does it signify?

Also, many other samples have no or negligible error bar, which cannot be true for
three biological replicates. The authors did not mention technical replicates.
Tubulin primers used in the current study are new or from the previous study? If
from the previous study needs a reference.

6. PLOS authors have the option to publish the peer
review history of their article (what does this mean?). If published, this will
include your full peer review and any attached files.

If you choose “no”, your identity will remain anonymous but your review may still be
made public.

**Do you want your identity to be public for this peer review?** For
information about this choice, including consent withdrawal, please see our
Privacy Policy.

Reviewer #1: No

Reviewer #2: Yes: Ritesh Kumar

---

## [Author Response · Author response to Decision Letter 0]

28 Jun 2020

Reviewer #1: 1. The author said that these genes were not only involved in Mg uptake
and transport, but also participated in Mg allocation among banana’s root, pseudo
stem, corm, and leaf components，which required more biological experiments to
support it

Response： the original sentences“The contrasting expression pattern of different Mg
transporter genes among differing tissues under Mg deficiency (Fig. 8) indicated
these genes were not only involved in Mg uptake and transport, but also participated
in Mg allocation among banana’s root, pseudo stem, corm, and leaf components”in page
7 line 21 - 24 changed to:

“The contrasting expression pattern of different Mg transporter genes among differing
tissues under Mg deficiency (Fig. 8) indicated these genes is probably not only
involved in Mg uptake and transport, but also participated in Mg allocation among
different tissues.”in page 7 line 13 – 16 in the revised version.

2. need adding bilogical stastics for q-PCR data

Response：We analyzed the significance of q-PCR data, as shown in Figure 8.

3. DISCUSSION should be rewritten.

Response: We changed the 1-8 paragraphs in the original DISCUSSION to the 1-3
paragraphs in the revised version.

4. writting and orgnization should be further improved.

Response: We carefully read the manuscript and improved the writting in the revised
version.

In the original version in page 1 line 10 – 11: “and the CorA/MGT/MRS2 family
proteins are considered vital to this process.” was change to: “especially
CorA/MGT/MRS2 family proteins, played a vital role in regulating Mg content in plant
cells” in page 1 line 10 -11;

In the original version in page 1 line 12 – 13: “much less is known about it for
tropical crops.” was change to: “the relevant information is scarce in tropical
crops.” in page 1 line 11 - 12;

In the original version in page 1 line 15: “chromosomes” was change to: “chromosome”
in page 1 line 15;

In the original version in page 1 line 18 – 19: “branches of the phylogenetic tree”
was change to: “phylogenetic” in page 1 line 18;

In the original version in page 1 line 22 – 28: “Specifically, among the MaMRS2
genes, MaMRS2-2 was expressed only in the corm, and MaMRS2-6 was most expressed in
the leaves. This result was confirmed by real-time PCR analysis which uncovered
differential MaMRS2 gene expression under Mg2+ deficiency conditions, in that
MaMRS2-5 and MaMRS2-7 were both up-regulated in stems, whereas MaMRS2-1 and
MaMRS2-10 were down-regulated in roots, stems, and leaves. These results will
contribute to the further study of Mg transporters in banana.”

was change to:

“The result was confirmed by real-time PCR analysis, in addition to tissue specific
expression, expression differences among MaMRS2 members was also observed under Mg
deficiency conditions. These results showed that Mg transporters may play a
versatile role in banana growth and development, and our work will shed light on the
functional analysis of Mg transporters in banana.” in page 1 line 21 - 26;

In the original version in page 2 line 1 - 4 “the concentration of Mg in metabolic
pools, such as the cytoplasm and chloroplast, are strictly regulated, for which
magnesium transporter (MGT) plays a vital role in maintaining the equilibrium and
homeostasis of Mg2+ in plants [6, 7].” changed to “The concentration of Mg in
metabolic pools, such as the cytoplasm and chloroplast, are strictly regulated. And
magnesium transporter (MGT) plays a vital role in maintaining the equilibrium and
homeostasis of Mg in plants [6, 7].” in page 1 line 37 – 38 and page 2 line 1 -2 in
the revised version.

In the original version in page 5 line 24 - 29 “was” changed to “were” in page 5 line
17 – 19 in the revised version.

Reviewer #2: The authors characterized the Magnesium transporter MaMRS2 protein
family. 10 MaMRS2 genes in banana (Musa acuminata) were identified. The
physicochemical properties, location on chromosomes, gene structure, cis-acting
elements, and replication relationships between these ten members were analyzed. The
tissue-specific expression pattern was analyzed. Three genes MaMRS2-1, MaMRS2-4, and
MaMRS2-7 were cloned and complemented with triple MGT mutant of Salmonella
typhimurium. This study could help in understanding the function of each MaMRS2 gene
in development or stress conditions.

The manuscript is poorly written and needs to be largely reworked.

Response：the original part in page 1 line 13-29 “In this study, 10 MaMRS2 genes in
banana (Musa acuminata) were isolated from its sequenced genome and classified into
five distinct clades. The physicochemical properties, location on chromosomes, gene
structure, cis-acting elements, and duplication relationships between these 10
members were analyzed. Complementary experiments revealed that three MaMRS2 gene
members (MaMRS2-1, MaMRS2-4, MaMRS2-7), from three distinct branches of the
phylogenetic tree, were capable of restoring the function of Mg2+ transport in
Salmonella typhimurium mutants. Semi-quantitative RT-PCR showed that the 10 members
of the MaMRS2 gene were differentially expressed in the root, pseudo stem, corm, and
leaves of banana cultivar ‘Baxijiao’ (Musa spp. AAA Cavendish) seedlings.
Specifically, among the MaMRS2 genes, MaMRS2-2 was expressed only in the corm, and
MaMRS2-6 was most expressed in the leaves. This result was confirmed by real-time
PCR analysis which uncovered differential MaMRS2 gene expression under Mg2+
deficiency conditions, in that MaMRS2-5 and MaMRS2-7 were both up-regulated in
stems, whereas MaMRS2-1 and MaMRS2-10 were down-regulated in roots, stems, and
leaves. These results should be helpful to further research work on Mg transporters
in banana crops.” changed to:

“In this study, 10 MaMRS2 genes in banana (Musa acuminata) were isolated from its
genome and classified into five distinct clades. The putative physicochemical
properties, chromosome location, gene structure, cis-acting elements, and
duplication relationships in between these members were analyzed. Complementary
experiments revealed that three MaMRS2 gene members (MaMRS2-1, MaMRS2-4, MaMRS2-7),
from three distinct phylogenetic branches, were capable of restoring the function of
Mg transport in Salmonella typhimurium mutants. Semi-quantitative RT-PCR showed that
MaMRS2 genes were differentially expressed in banana cultivar ‘Baxijiao’ (Musa spp.
AAA Cavendish) seedlings. The result was confirmed by real-time PCR analysis, in
addition to tissue specific expression, expression differences among MaMRS2 members
was also observed under Mg deficiency conditions. These results showed that Mg
transporters may play a versatile role in banana growth and development, and our
work will shed light on the functional analysis of Mg transporters in banana.” In
page 1 line 13 -29 in the revised version

1. Rewrite the line: These results should be helpful to further research work on Mg
transporters in banana crops.

Response：the original “These results should be helpful to further research work on Mg
transporters in banana crops” in page 1 line 27 - 29 changed to:

“These results showed that Mg transporters may play a versatile role in banana growth
and development, and our work will shed light on the functional analysis of Mg
transporters in banana.” in page 1, line 24-26 in the revised version.

2. The introduction is very messy, especially the expression summary of MGTs in
different plants. Rewrite the introduction.

Response: we have rewritten the INTRODUCTION.

The original sentence in page 1 line 32 – 33 “ During plant growth and development,
magnesium (Mg) is essential and it cannot be substituted[1]” changed to：

“Magnesium (Mg) is essential in plant growth and development, and cannot be
substituted[1].” In page 1 line 30 – 31 in the revised version;

The original sentence in page 1 line 34-36 “The major function of Mg in green leaves
is forming the central atom of chlorophyll, Mg is also involved in protein synthesis
as a bridging element for the aggregation of ribosome subunits [2], and functions as
a stabilizer of specific conformation in nucleic acid synthesis [3]” changed to: “Mg
is also involved in protein and nucleic acid synthesis, and it acts as a bridging
element for the aggregation of ribosome subunits [2] and a conformation stabilizer
[3] respectively.” In page 1 line 32 - 34 in the revised version;

The original sentence in page 2 line 1 – 4 “The concentration of Mg in metabolic
pools, such as the cytoplasm and chloroplast, are strictly regulated, for which
magnesium transporter (MGT) plays a vital role in maintaining the equilibrium and
homeostasis of Mg2+ in plants [6, 7].” changed to: “The concentration of Mg in
metabolic pools, such as the cytoplasm and chloroplast, are strictly regulated. And
magnesium transporter (MGT) plays a vital role in maintaining the equilibrium and
homeostasis of Mg in plants [6, 7].” In page 1 line 37 – 38 and page 2 line 1 – 2 in
the revised version;

The expression summary of MGTs in different plants in the original part in page 2
second and third paragraphs changed to page 2 second paragraph in the revised
version；

The original sentence in page 3 line 6 – 10 “Imbalanced fertilization practices
aggravates the likelihood of Mg deficiency occurring in cultivated banana, such that
Mg deficiency is now a major reason for reduced banana fruit yield [30]. Therefore,
improving the efficiency of Mg utilization in banana has immediate practical
significance. Clearly, MGT plays a vital role in Mg nutrition maintence in plant
species.” changed to: “Imbalanced fertilization practices aggravates the likelihood
of Mg deficiency, as a result, Mg deficiency is now a major contributor to banana
yield reduction [30]. Therefore, improving the efficiency of Mg utilization in
banana has immediate practical significance. Clearly, MGT plays a vital role in Mg
nutrition maintenance in plants.” in page 2 line 41 – 44 and page 3 line 1 in the
revised version;

3. Units are missing: The isoelectric point of the predicted proteins and their
corresponding amino acid length ranged respectively from 4.51 to 9.16 (pI) and from
379 to 495 (aa).

Response: We have supplemented the corresponding units “Identification of MRS2 family
genes in banana” ,the isoelectric point, the amino acid length and the molecular
weight of the predicted proteins ranged from 4.51 to 9.16 (pI), from 379 to 495
(aa), and from 42.57 to 54.83 (kDa)” in Page 3 line 15 in the revised version.

4. Rewrite: Figure 2 shows the phylogenetic tree built for Mg transporter protein
sequences from banana, Arabidopsis, rice, maize, and yeast, with yeast Mg
transporters as the outgroup.

Response：the original sentence in page 5 line 4 - 6 “Figure 2 shows the phylogenetic
tree built for Mg transporter protein sequences from banana, Arabidopsis, rice,
maize, and yeast, with yeast Mg transporters as the outgroup” changed to:

“Phylogenetic trees were constructed using MGTs of banana, Arabidopsis, rice, maize
and yeast. Among them, MGTs of yeast was selected as the outgroup (Fig. 2)”. In Page
5, line 3 – 4 in the revised version.

5. Chromosomal location and gene duplication: Authors mentioned gene duplication as
gene replication. (5 times). There is a difference between the two terms.

Also, in the discussion part: we found evidence for replication relationships that
occurred between MaMRS2 family members across chromosomes but no signs of a tandem
replication relationship (Fig. 4). This pattern generally exists among other gene
family members of banana [32]. It is reasonable to speculate that multiple rounds of
whole genome replication events have occurred over evolutionary time in banana.

Response: we modified “replication” in page 5 line 39 = 42 and page 6 line 1 to
“duplication” in page 5 line 37 – 42 (5 times) in the revised version.

Also, the original sentence “This pattern generally exists among other gene family
members of banana [32]. It is reasonable to speculate that multiple rounds of whole
genome replication events have occurred over evolutionary time in banana.” in page 8
line 14 -16 changed to:

“Genomic collinearity showed that duplication relationships occurred between MaMRS2
family members across chromosomes, but no signs of a tandem duplication relationship
(Fig. 4), a pattern generally exists among other gene family members of banana” in
the revised version.

6. Figure legends are not uploaded properly or incomplete:

Fig. 1 Multiple sequence alignments of MaMSR2 proteins. This alignment was performed
using DNAMAN software. The identical, conserved, and less conserved

Fig. 2 Phylogenetic analysis of Arabidopsis, rice, maize, yeast, and banana
CorA/MRS2/MGT members. The neighbor-joining tree, which includes 10 MaMRS2

Fig. 3 Phylogenetic relationships (A), gene structure (B), and motif analysis (C) of
MaMRS2 family members in banana. The phylogenetic tree was constructed using the

Fig. 4 Chromosomal location and gene duplication of MaMRS2 genes in the banana
genome. The MaMRS2 gene is located on multiple chromosomes. The chromosome

Fig. 5 Predicted cis-acting elements in the promoter region of the 10 MaMRS2 genes in
banana. Different colors represent cis-acting elements associated with different

Fig. 6 Complementary analysis of the MaMRS2 genes. MM1927 is the wild type, used here
as a positive control; the MM281 and

Fig. 7 Expression of 10 MaMRS2 genes in different tissues of banana cultivar
‘Baxijiao’ (Musa spp. AAA Cavendish) seedlings. Explain the Abbreviations.

Fig. 8 Relative expression levels of 10 MaMRS2 genes in different tissues of banana
cultivar ‘Baxijiao’ (Musa spp. AAA Cavendish) Explain the Abbreviations.

Response: We have uploaded the complete Figure legends.

Fig.1 Multiple sequence alignments of MaMSR2 proteins. Alignment was performed using
DNAMAN software. The identical, conserved and less conserved amino acid residues are
indicated by dark, cherry red and cyan background colors, respectively. The
conservative GMN motif was indicated and the TM domains are underlined.

Fig.2 Phylogenetic analysis of Arabidopsis, rice, maize, yeast and banana
CorA/MRS2/MGT members. The Neighbor-Joining tree, which includes 10 MaMRS2 protein
from banana, 11 MRS2/MGT proteins from Arabidopsis, 9 MRS2/MGT proteins from rice,
12 MRS2/MGT proteins from maize and 5 MRS2/MGT proteins from yeast, was constructed
using MEGA X. A, B, C, D, E, F, G and H represent the different clades of the
MRS2/MGT family in these five species.

Fig.3 Phylogenetic relationships (A), gene structure (B) and motif analysis (C) of
MaMRS2 family members. The Phylogenetic tree was constructed using the
Neighbor-Joining method with 1000 bootstrap replicates in the MEGA X software. Then
the gene structure was performed using GSDS program. MEME program and TBTools were
used to illustrate the motif analysis results. Yellow boxes and black lines
represent exons and introns, respectively, blue boxes indicate 5’ or 3’ untranslated
regions (UTRs) and different motif was painted with different color.

Fig.4 Chromosomal location and gene duplication of MaMRS2 genes in the banana genome.
The MaMRS2 gene is located on different chromosomes. The number of chromosomes is
shown on the outside, and the different colors represent different chromosomes. The
grey region is the collinearity of the banana genome, highlighting the collinearity
between the MaMRS2 genes was highlighted with red lines.

Fig.5 Analysis of cis-acting elements of members of the MaMRS2 gene family. Different
colors represent cis-acting elements of different functions.

Fig.6 Complementary analysis of the MaMRS2 genes. MM1927 was the wild type and used
as a positive control, MM281 and MM281 transformed with an empty pTrc99A were used
as negative controls. (A) Functional verification on N minimal solid medium
containing 20, 10, 5, 2, 1, 0.5, 0.1, 0.01 mM MgSO4. The bacterial was diluted
sequentially 10-fold from left to right. (B) Growth curves were performed in
N-minimal liquid medium containing 10, 1, 0.5 and 0.01mM MgSO4, and the cell density
was monitored at OD600 every 2 hours. Data was an average of three independent
experiments, and the different icons in the figure represent the average of three
repetitions.

Fig.7 Expression of MaMRS2 gene in different tissues of Baxijiao seedlings. UL, LL,
PS, C and R represent upper leaf, lower leaf, pseudo stem, corm and root
respectively.

 Fig.8 Relative expression level of MaMRS2 gene in different tissues of Baxijiao
seedlings under magnesium deficiency. The CK indicated the value under normal growth
conditions as a control, and the -Mg indicated the relative expression level under
the complete absence of magnesium ion condition. UL, LL, PS, C and R represent upper
leaf, lower leaf, pseudo stem, corm and root respectively.

7. The author mentioned sequences of 7 genes:

Supplementary Test1 The nucleotide sequences of the seven MaMRS2 genes from
sequencing

There are only three sequences >MaMRS2-1, >MaMRS2-4, >MaMRS2-7 as authors
clones only three genes.

Response: The original “Supplementary Test1 The nucleotide sequences of the seven
MaMRS2 genes from sequencing” changed to “Supplementary Test1 The nucleotide
sequences of the three MaMRS2 genes from sequencing” see Supplementary test1.

8. The authors did complementation assay with MM281 mutant.

Figure 6A: Why the growth of complemented lines is low in higher concentration of
magnesium as compare to Mutant (MaMRS2-4, MaMRS2-7), as Complemented lines are
growing fine in lower concentrations.

Response: We speculate that the growth of the transformants has an optimal
concentration. the genes from banana may be less efficient under high Mg condition
when compared with that from bacteria, while they are similar under low Mg
conditions.

Figure 6B: Authors mentioned: The data are shown as the mean ± SD of three biological
replicates,

However, no SD is shown on the line charts.

Response: “The data are shown as the mean ± SD of three biological replicates,” in
page 12 line 19 changed to “The data are shown as the mean of three biological
replicates,” in page 11 line 4 in the revised version.

The growth curve at 0.01 mM Mg2+ shows that complementing lines 2-1 and 2-7 are
growing better than the wild type, which does not correlate with Figure 6A.

Response: We carefully checked the original data and found that there were errors in
the data processing. Now we have corrected the original data. Figure 6B 0.01mm is
corrected (please see the revised figure 6B):

9. qRT-PCR analysis of MaMRS2-9 has no error bars; what does it signify? Also, many
other samples have no or negligible error bar, which cannot be true for three
biological replicates. The authors did not mention technical replicates.

Response: The histogram in Figure 8 is the average of three technical repetitions
with 3 biological replicates, we revised it in the new manuscript with error bars
added.

Tubulin primers used in the current study are new or from the previous study? If from
the previous study needs a reference.

Response: Tubulin primers used in the current study are from the previous study, The
reference was in Page 12, line 29 - 31 “The expression level of each MaMRS2 gene was
calculated using the 2-ΔCt method, for which the TUB gene served as the internal
reference [56]”. See Reference: 56. Podevin N, Krauss A, Henry I, Swennen R, Remy S.
Selection and validation of reference genes for quantitative RT-PCR expression
studies of the non-model crop Musa. Mol Breeding. 30(3): 1237-1252.https://doi.org/10.1007/s11032-012-9711-1 PMID:
23024595.

Also, we carefully read our original article and found some mistakes, as follows:

the original sentences in page 6 line 2“The Ka/Ks of five homologous Mg transporters
was < 0.3 (Supplementary Table S1);” have changed to “The Ka/Ks of five
homologous Mg transporters was less than 0.3 (Supplementary Table S1);”in page 5
line 43-44 in the revised version.

to Reviewers.docx
---

## [Decision Letter · Decision Letter 1]

31 Jul 2020

PONE-D-20-02641R1

Identification and functional analysis of the CorA/MGT/MRS2-type magnesium
transporter in banana

PLOS ONE

Dear Dr. Huang,

Thank you for submitting your manuscript to PLOS ONE. After careful consideration, we
feel that it has merit but does not fully meet PLOS ONE’s publication criteria as it
currently stands. Therefore, we invite you to submit a revised version of the
manuscript that addresses the points raised during the review process.

Please incorporate suggestions of reviewer 2 in the revised manuscript.

Please submit your revised manuscript by Sep 14 2020 11:59PM. If you will need more
time than this to complete your revisions, please reply to this message or contact
the journal office at plosone@plos.org. When
you're ready to submit your revision, log on to https://www.editorialmanager.com/pone/ and select the 'Submissions
Needing Revision' folder to locate your manuscript file.

If you would like to make changes to your financial disclosure, please include your
updated statement in your cover letter. Guidelines for resubmitting your figure
files are available below the reviewer comments at the end of this letter.

We look forward to receiving your revised manuscript.

Kind regards,

Anil Kumar Singh, Ph.D.

Academic Editor

PLOS ONE

Reviewers' comments:

Reviewer's Responses to Questions

**Comments to the Author**

1. If the authors have adequately addressed your comments raised in a previous round
of review and you feel that this manuscript is now acceptable for publication, you
may indicate that here to bypass the “Comments to the Author” section, enter your
conflict of interest statement in the “Confidential to Editor” section, and submit
your "Accept" recommendation.

Reviewer #1: All comments have been addressed

Reviewer #2: All comments have been addressed

2. Is the manuscript technically sound, and do the data
support the conclusions?

Reviewer #1: Yes

Reviewer #2: Yes

3. Has the statistical analysis been performed
appropriately and rigorously? 

Reviewer #1: Yes

Reviewer #2: Yes

4. Have the authors made all data underlying the
findings in their manuscript fully available?

Reviewer #1: Yes

Reviewer #2: Yes

5. Is the manuscript presented in an intelligible
fashion and written in standard English?

Reviewer #1: Yes

Reviewer #2: Yes

6. Review Comments to the Author

Reviewer #1: (No Response)

Reviewer #2: Authors have addressed all the comments and quality of manuscript is now
acceptable. Authors may include this response in Discussion:

Comment8. The authors did complementation assay with MM281 mutant.

Figure 6A: Why the growth of complemented lines is low in higher concentration of

magnesium as compare to Mutant (MaMRS2-4, MaMRS2-7), as Complemented lines

are growing fine in lower concentrations.

Response: We speculate that the growth of the transformants has an optimal

concentration. the genes from banana may be less efficient under high Mg
condition

when compared with that from bacteria, while they are similar under low Mg

conditions.

English quality is much better than the first version.

Also statistical analysis is now clear and acceptable.

7. PLOS authors have the option to publish the peer
review history of their article (what does this mean?). If published, this will
include your full peer review and any attached files.

If you choose “no”, your identity will remain anonymous but your review may still be
made public.

**Do you want your identity to be public for this peer review?** For
information about this choice, including consent withdrawal, please see our
Privacy Policy.

Reviewer #1: No

Reviewer #2: **Yes: **Ritesh Kumar

---

## [Author Response · Author response to Decision Letter 1]

12 Aug 2020

Reviewer #2:

1.Authors may include this response in Discussion:

Comment8. The authors did complementation assay with MM281 mutant.

Figure 6A: Why the growth of complemented lines is low in higher concentration of

magnesium as compare to Mutant (MaMRS2-4, MaMRS2-7), as Complemented lines

are growing fine in lower concentrations.

Response:

In the reviewed manuscript at the end of the second paragraph of the discussion, we
added the following:

The complementary lines incorporated with MaMRS2-4 and MaMRS2-7 could grow well under
low Mg concentration conditions, but they had lower growth rate when compared with
the mutant control under solid growth medium with higher Mg concentration (10 mM and
20 mM) (Fig. 6A). These results indicated that both MaMRS2-4 and MaMRS2-7could
transport Mg, at the same time, the exogenous Mg transporters from banana might play
its role in excessive Mg accumulation and lead to the toxic effect in bacteria under
high Mg conditions.

2.Also, we carefully read our original article and found some mistakes, as
follows:

The “physicochemical” was changed to “physiochemical” in page 1 line 15；

“was” was changed were” in page 1 line 23;

“isocitrate-lyase” was changed “isocitrate lyase” page 1 line 35;

“RuBP carboxylase” was changed “ribulose bisphosphate carboxylase” in page 2 line
36;

“diversed” was changed “diverse” in page 2 line 35;

“Protiens” was changed “proteins” in page 9 line 6;

“Figure. 6 A” was changed “Fig. 6” in page 9 line 7;

“Ca(NO3)·4H2O” was changed “Ca(NO3)2·4H2O” in page 11 line 20

“Chen C, Xia R, Chen H, He Y. TBtools, a Toolkit for Biologists integrating various
HTS-data handling tools with a user-friendly interface. bioRxiv. 2018: 289660.
https://doi.org/10.1101/289660.” was changed to
“Chen C, Chen H, Zhang Y, et al. TBtools-an integrative toolkit developed for
interactive analyses of big biological data[J]. bioRxiv, 2020: 289660. https://doi.org/10.1016/j.molp.2020.06.009” in
page 16 line 15-17.

to Reviewers.docx
---

## [Editor Report · Decision Letter 2]

31 Aug 2020

Identification and functional analysis of the CorA/MGT/MRS2-type magnesium
transporter in banana

PONE-D-20-02641R2

Dear Dr. Huang,

We’re pleased to inform you that your manuscript has been judged scientifically
suitable for publication and will be formally accepted for publication once it meets
all outstanding technical requirements.

Kind regards,

Anil Kumar Singh, Ph.D.

Academic Editor

PLOS ONE
---

## [Editor Report · Acceptance letter]

17 Sep 2020

PONE-D-20-02641R2

Identification and functional analysis of the CorA/MGT/MRS2-type magnesium
transporter in banana

Dear Dr. Huang:

I'm pleased to inform you that your manuscript has been deemed suitable for
publication in PLOS ONE. Congratulations! Your manuscript is now with our production
department.

Kind regards,

on behalf of

Dr. Anil Kumar Singh 

Academic Editor

PLOS ONE